# Cyclic Change in Right and Left Ventricular Systolic and Diastolic Function in Patients with Neuromuscular Disorders on Permanent Mechanical Ventilation

**DOI:** 10.3390/jpm12101694

**Published:** 2022-10-11

**Authors:** Abdallah Fayssoil, Nicolas Mansencal, Jean Bergounioux, Karim Wahbi, Tanya Stojkovic

**Affiliations:** 1EchoLab, Raymond Poincaré Hospital, Université de Versailles-Saint Quentin, 78180 Montigny-le-Bretonneux, France; 2University of Paris-Saclay, INSERM U1179, END-ICAP, 91190 Versailles, France; 3Cardiology Department, Ambroise Paré Hospital, APHP, UVSQ, INSERM U1018, Centre de Référence des Cardiomyopathies, 92100 Boulogne Billancourt, France; 4APHP, Pitié-Salpêtrière Hôpital Reference, Center for Neuromuscular Diseases, Institute of Myology, 75013 Paris, France; 5Pediatric Intensive Care Unit, Raymond Poincaré Hospital, APHP, 92380 Garches, France; 6Cardiology Unit, Cochin Hospital, APHP, 75013 Paris, France

**Keywords:** Duchenne muscular dystrophy, congenital myopathy, echocardiography, ventilation

## Abstract

Home mechanical ventilation is classically used to treat neuromuscular patients with chronic respiratory insufficiency. Since the heart and lungs are localized in the thorax, intrathoracic pressure may directly affect heart function. Here, we report the direct cyclic effects of mechanical ventilation on the right and left ventricular systolic and diastolic function in serial cases. These cases highlight the importance of performing Doppler echocardiography in patients with neuromuscular disorders, focusing on the right and left ventricular systolic and diastolic functions in the context of ventilation setting modifications or introduction.

## 1. Introduction

Neuromuscular disorders are acquired or inherited (genetic) conditions that can affect either motor neurons, the peripheral nervous system, skeletal muscles, or neuromuscular junction. Among the hereditary myopathies, Duchenne muscle dystrophy (DMD) represents the most frequent hereditary childhood myopathy, affecting approximately 19.8 in 100,000 male births worldwide [1]. Respiratory function can be affected, as well as cardiac function, in patients with neuromuscular disorders, particularly in patients suffering from DMD. Home mechanical ventilation (HMV) is classically used to treat neuromuscular patients with chronic respiratory insufficiency, nocturnal hypoventilation, and apnea hypopnea syndrome [2]. Because the heart and lungs are localized in the chest cavity, intrathoracic pressure may directly affect heart function, whether the patient is spontaneously breathing or on mechanical ventilation. The heart–lung relationship becomes more significant for a patient in the context of mechanical ventilation, neuromuscular disorders, chest deformities, altered lung compliance, and hypovolemia. Here, we report a case series of cyclic changes in the right and left ventricular systolic and diastolic function in patients with hereditary myopathy who are on permanent mechanical ventilation. This study followed the ethical principles formulated in the Declaration of Helsinki. Informed consent was obtained from participants included in this study.

### 1.1. Case 1

A 29-year-old male DMD was admitted for routine cardiac evaluation in the EchoLab. The patient was wheelchair-bound, received angiotensin converting enzyme (ACE) inhibitor because of previous left ventricular systolic dysfunction (left ventricular ejection fraction (LVEF) at 45%), and was hemodynamically stable. He was invasively ventilated because of severe respiratory insufficiency (forced predictive pulmonary vital capacity (VC) at 8%). Mechanical ventilation was performed 24 h/24 h using the following parameters: tidal volume = 400 mL and positive end-expiratory pressure (PEEP) = 6 cmH_2_O. Current transthoracic echocardiography disclosed recovered left ventricular systolic function. With mechanical ventilation, and according to insufflation, we noted modifications of the cardiac dimensions and volumes. Indeed, during the insufflation phase, we observed an increase in the LVEF, varying from 63% to 76%, a reduction in the meantime of the left ventricular end-diastolic diameter (LVEDD), from 40 mm to 32 mm, and a reduction of the end-diastolic ventricular volume (from 71 mL to 41 mL during the insufflation phase) due to the intermittent increase in the intrathoracic pressures (Figure 1).

### 1.2. Case 2

A 27-year-old male DMD was admitted to the EchoLab for an annual cardiac evaluation. He was invasively ventilated 24 h/24 h (tidal volume 480 mL, PEEP 3 cm H_2_O) because of severe respiratory insufficiency (forced predictive pulmonary VC at 6%). He took a cardioprotective drug, namely an ACE inhibitor. His past echocardiography disclosed normal left ventricular systolic function. At admission, systolic and diastolic blood pressures were respectively 110 mmHg and 59 mmHg. The electrocardiogram revealed a sinus rhythm with an incomplete right bundle branch block. Transthoracic echocardiography disclosed preserved LVEF (65%) and normal LVEDD (36 mm). We found a cyclic hemodynamic variation with a cyclic reduction of the left ventricular outflow tract flow peak velocity and a reduction of the aortic time during the insufflation phase (from 258 ms to 207 ms) (Figure 2). 

### 1.3. Case 3

A 29-year-old female patient with congenital myopathy was admitted to the EchoLab for a cardiac evaluation. The patient was invasively ventilated with permanent mechanical ventilation (24 h/24 h). She was wheelchair-bound and had a gastrostomy. Ventilation setting parameters were as follow: tidal volume 480 mL, PEEP 5 cm H_2_O, and a respiratory rate of the ventilator at 16. She took no medications. The electrocardiogram (ECG) revealed a sinus rhythm with an incomplete left bundle-branch block. Predictive pulmonary VC was at 6%. Transthoracic echocardiography revealed a left ventricular systolic dysfunction with an LVEF at 40%, as well as a right ventricular (RV) systolic dysfunction attested by an RV annular tissue Doppler peak velocity at 8 cm/s. We found a cyclic RV systolic and diastolic dysfunction with a decrease in the RV tissue Doppler peak velocity during insufflation. The peak systolic tissue Doppler RV velocity decreased from 8 cm/s to 5 cm/s during insufflation, and the peak early diastolic tissular Doppler RV velocity decreased from 11 cm/s to 6 cm/s (Figure 3). The ratio peak early inflow tricuspid velocity/peak early tissue RV Doppler velocity (tric E/e’) increased from 3.5 to 6 during the insufflation phase, illustrating an RV diastolic impairment during the insufflation phase (Figure 3). 

### 1.4. Case 4

A 29-year-old male patient suffering from congenital myopathy was admitted for cardiac evaluation in the EchoLab. He was invasively ventilated 24 h/24 h (tidal volume 450 mL and PEEP 2 cm H_2_O) because of severe chronic respiratory insufficiency. His past medical history was pertinent for gastrostomy and thoracic drainage because of pneumothorax. Body mass index (BMI) was 36 kg/m². Systolic and diastolic blood pressures were, respectively, 118 mmHg and 70 mmHg, and the heart rate was 95 bpm. Results of the blood gas exchange were as follow: pH = 7.47, PCO2 = 29 mmHg, PO2 = 100 mmHg, and bicarbonates at 21 mmol/L. The ECG revealed a sinus rhythm with an incomplete right bundle-branch block. The echocardiography found normal LVEF (70%) and normal right ventricular systolic function. We found a cyclic diastolic tricuspid filling pattern variation with a peak early diastolic inflow tricuspid velocity (E) of 57 cm/s, decreasing to 36 cm/s during the insufflation phase, whereas the peak atrial velocity of the tricuspid inflow (A) changed from 53 cm/s to 42 cm/s during insufflation (Figure 4).

### 1.5. Case 5

A 51-year-old male patient suffering from rod congenital myopathy was admitted for cardiac evaluation in the EchoLab. He was invasively ventilated 20 h/24 h (tidal volume 500 mL and PEEP 5 cm H_2_O) because of severe chronic respiratory insufficiency. The patient was wheelchair-bound, and BMI was 23 kg/m². Systolic and diastolic blood pressures were, respectively, 140 mmHg and 80 mmHg, and the heart rate was 95 bpm. NT-proBNP was 54 ng/L. The ECG revealed a sinus rhythm. The echocardiography found normal systolic ventricular function (LVEF = 65%). The RV systolic function was preserved with a right fractional area change (FAC) at 55%. We found a cyclic RV systolic and diastolic dysfunction with a decrease in the RV annular tissue Doppler peak velocity during insufflation. The peak systolic tissue Doppler RV velocity decreased from 12 cm/s to 8 cm/s during insufflation, and the peak early diastolic tissular Doppler RV velocity decreased from 8 cm/s to 6 cm/s. We found, in addition, a cyclic diastolic tricuspid filling pattern with a peak early diastolic inflow tricuspid velocity (E) at 66 cm/s, decreasing to 50 cm/s during the insufflation phase, whereas the peak atrial velocity of the tricuspid inflow (A) decreased from 58 cm/s to 49 cm/s. The ratio peak early inflow tricuspid velocity/peak early tissular RV Doppler velocity (tric E/e’) was increased (tric E/e’ 8) and was stabilized during insufflation (tric E/e’ from 8 without insufflation to 8 during insufflation), illustrating the direct impact of the mechanical ventilation on venous return. Table 1 summarizes the Doppler echocardiographic data of the five patients included in the study.

## 2. Discussion

In this case series highlighting different types of hereditary myopathies and including five patients requiring invasive ventilation, we found that (i) mechanical ventilation may induce a cyclic improvement of the left systolic ventricular function associated with a cyclic reduction of the left ventricular end-diastolic volumes, an index of preload, as shown in the first case; (ii) mechanical ventilation may impede the left ventricular hemodynamics with a cyclic reduction of left ventricular outflow peak velocity, aortic time, and stroke volume, as noticed in cases 1 and 2; and (iii) finally, it may alter the right ventricle systolic function and subsequently the cyclic right ventricular systolic dysfunction. It can also affect the right ventricular diastolic filling pattern in a cyclic manner (cases 3–5).

Positive-pressure ventilation increases intrathoracic pressure. Since diaphragmatic descent increases intra-abdominal pressure, the decrease in the pressure gradient for venous return is less than would otherwise occur if the only change were an increase in right atrial pressure. However, in hypovolemic states, it can induce profound decreases in venous return. Increases in intrathoracic pressure decrease left ventricular afterload and will augment left ventricular ejection. In patients with hypervolemic heart failure, this afterload-reducing effect can result in improved left ventricular ejection, increased cardiac output, and reduced myocardial O2 demand. These results illustrate the key role of the venous return and the intermittent cyclic change of hemodynamic status due to heart–lung interaction in the context of mechanical ventilation in patients with neuromuscular disorders and the tremendous emphasis of this physiological mechanism by specific patient condition. Venous return depends on the mean systemic pressure and the right atrial pressure, that is, the backpressure [3]. The mean systemic pressure is determined by peripheral vasomotor tone, blood volume, and blood-flow distribution [4]. Conversely with spontaneous breathing, the insufflation during mechanical ventilation increases the intrathoracic pressure (ITP) [4]. High ITP results in an increase in the right atrial pressure that impedes the venous return to the RV [5]. In this context, mechanical ventilation may induce a reduction of the cardiac output [6]. In DMD patients invasively, permanently ventilated because of severe chronic respiratory insufficiency, with the insufflation cycle, intrathoracic pressures, which impede the venous return, may decrease cardiac preload. Here, we have illustrated the immediate effects of the mechanical ventilation on the cardiac and hemodynamic function in this group of patients. Since, during the same exam, LVEF, cardiac diameters and volumes may vary, it is essential to provide the measurements at end of the expiratory phase and not during the insufflation cycle when monitoring patients. The right ventricular diastolic function may also be affected in patients with cardiomyopathies and in patients with respiratory diseases [6], with the ratio tricuspid E/e’ used to estimate the RV filling pressure [7]. 

For a clinical perspective and clinical practice, reducing tidal volume, reducing PEEP, and prolonging expiratory time may have a positive impact on the hemodynamic status of patients with neuromuscular disorders on mechanical ventilation [2]. Doppler echocardiography should be performed in patients with neuromuscular disorders, focusing on the right and left systolic and diastolic functions within the context of ventilation setting modifications or before the introduction of mechanical ventilation.

## Figures and Tables

**Figure 1 jpm-12-01694-f001:**
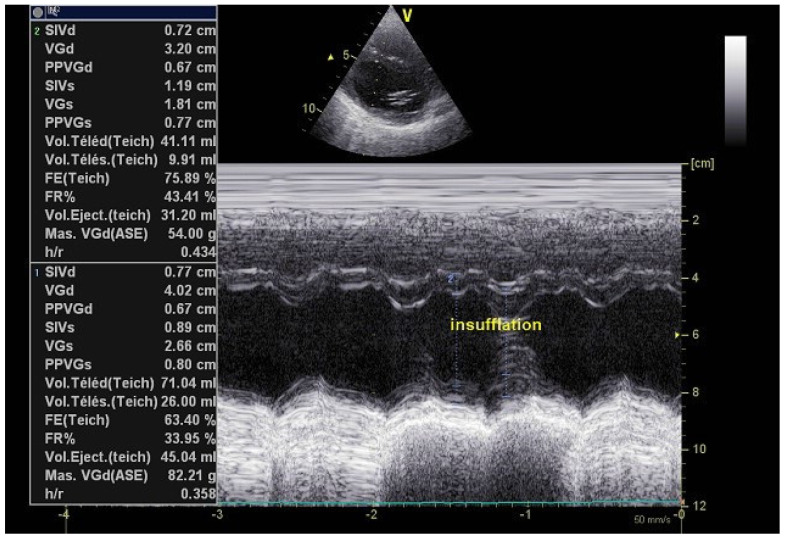
Left ventricular time movement ™ mode providing left ventricular dimensions in tracheotomized DMD patient on permanent HMV. 2: cardiac measurements during insufflation cycle; 1: cardiac measurement during non-insufflation phase. VGd: left ventricular end-diastolic diameter; FE: left ventricular ejection fraction; Vol.Teled: left ventricular end-diastolic volume; Vol. Télés: left ventricular end-systolic volume. HMV: home mechanical ventilation.

**Figure 2 jpm-12-01694-f002:**
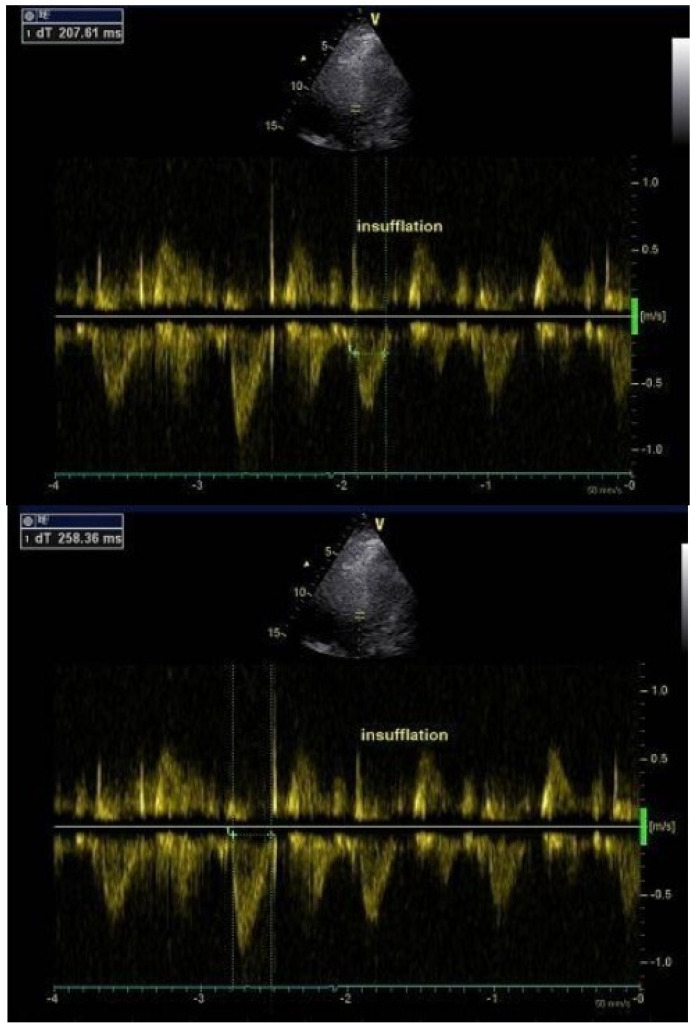
LVOT pulsed Doppler flow, with and without insufflation, in a tracheotomized DMD patient on permanent HMV. During the insufflation, note the reduction of the Doppler peak LVOT velocity and the reduction of the aortic ejection time (from 258 ms to 207 ms). DMD: Duchenne muscular dystrophy; LVOT: left ventricular outflow tract; HMV: home mechanical ventilation.

**Figure 3 jpm-12-01694-f003:**
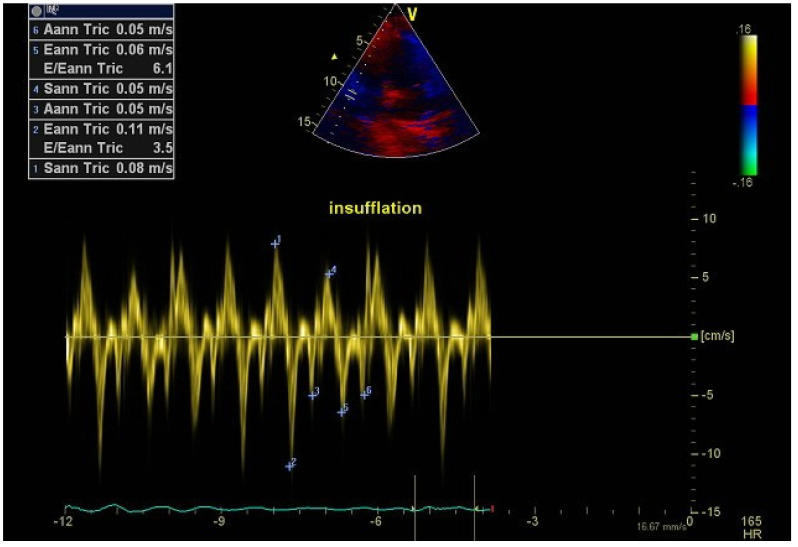
Right ventricular (RV) systolic and diastolic tissue Doppler peak velocities in a tracheotomized patient with congenital myopathy on permanent mechanical ventilation. Note the reduction of the systolic (S’) and diastolic (e’, a’) peak velocities during insufflation that disappear during the non-insufflation phase.

**Figure 4 jpm-12-01694-f004:**
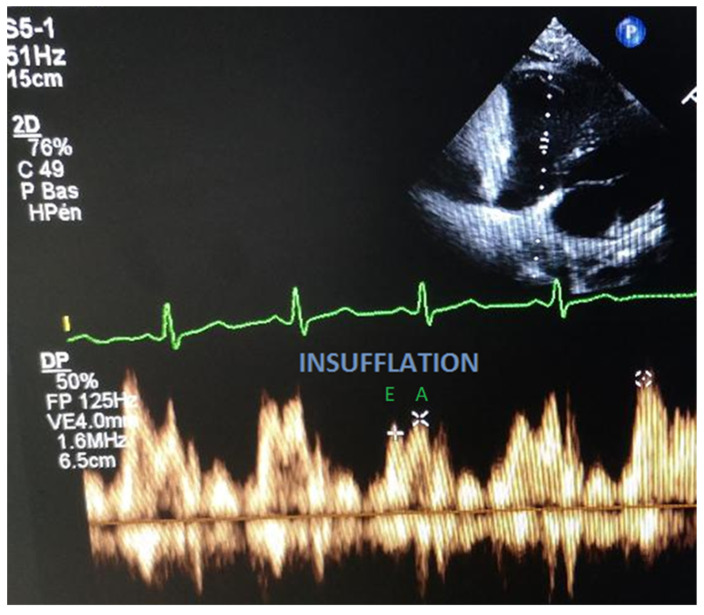
RV diastolic inflow pattern in a tracheotomized patient on permanent mechanical ventilation. We recorded the peak tricuspid early inflow velocity (E) and the peak atrial velocity (A). Note the reduction of peak E and peak A velocity during insufflation. RV: right ventricle.

**Table 1 jpm-12-01694-t001:** Doppler echocardiographic data of the five patients presenting with neuromuscular disorders on mechanical ventilation.

Parameters	Tidal Vol (mL)	PEEP (cmH20)	LVEDD (mm)	subAo VTI (cm)	LVEF1 (%)	LVEF2 (%)	RV S’1 (cm/s)	RV S’2 (cm/s)	RV e’ 1 (cm /s)	RV e’ 2 (cm/s)	RV a’ 1 (cm/s)	RV a’ 2 (cm/s)	Tric E 1 (cm/s)	Tric E 2 (cm/s)	Tric A 1 (cm/s)	Tric A 2 (cm/s)
Case 1	400	6	40	11	63	76										
Case 2	480	3	36	15	65	69										
Case 3	480	5	50	14	40	50	8	5	11	6	5	5	47	31	27	21
Case 4	450	2	49		70		16	12	18	16	15	15	57	36	53	42
Case 5	500	5		16	65		12	8	8	6	10	8	66	50	58	49

Measurements were recorded without insufflation (1) and during insufflation (2). LVEDD: left ventricular end-diastolic diameter; LVEF: left ventricular ejection fraction; PEEP: positive end-expiratory pressure; RV a’: peak late diastolic tissue Doppler right ventricular velocity; RV é: peak early diastolic tissue Doppler right ventricular velocity; RV S’: peak systolic tissue Doppler right ventricular velocity; SubAo VTI: subaortic velocity time integral; Tric A: peak atrial velocity from the right ventricular diastolic inflow pattern; Tric E: peak tricuspid early inflow velocity.

## Data Availability

The data presented in this study are available in the article.

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
