# Peer review of "Cyclic Change in Right and Left Ventricular Systolic and Diastolic Function in Patients with Neuromuscular Disorders on Permanent Mechanical Ventilation"

_jpm, 2022, doi:10.3390/jpm12101694_

Round 1

Reviewer 1 Report

In this review, FAYSSOIL et al. report the direct cyclic effects of mechanical ventilation on right and left ventricular systolic and diastolic function in DMD patients. I have several significant concerns.
Major Comments

1.     It is well known that DMD patients displayed cardiomyopathy in late life. Do you have the data showing heart defects in DMD patients?

2.     For each patient, can you provide all the five parameters (left ventricular dimensions, LVOT pulsed Doppler flow, Right ventricular (RV) systolic and diastolic tissue Doppler peak velocities, RV diastolic inflow pattern, Right ventricular (RV) systolic and diastolic tissue Doppler peak velocities).

3.     The manuscript is clear, but there are many grammatical errors. I suggests that editing for academic English is necessary and would significantly improve the communication of the findings of this paper.

Minor concerns:

1.     In Abstracts, “We report case series of direct cyclic effects of mechanical ventilation on right and left ventricular systolic and diastolic function.” should be changed into ” We report the direct cyclic effects of mechanical ventilation on right and left ventricular systolic and diastolic function in serial cases.”

2.     In Abstracts, “These cases highlight the importance of performing echocardiography Doppler in patients with neuromuscular disorders”should be changed into “These cases highlight the importance of performing Doppler echocardiography in patients with neuromuscular disorders”

3.     In the Introduction, “Neuromuscular disorders are a group of hereditary and acquired neurological diseases with skeletal disability.” should be changed into “Neuromuscular disorders are acquired or inherited (genetic) conditions that affect the peripheral nervous system and skeletal muscles.”

4.     In the Introduction, “Duchenne muscle dystrophy (DMD) represent the most frequent hereditary childhood myopathy (global DMD birth prevalence: 19.8/100,000 live male births)” should be changed into “Duchenne muscle dystrophy (DMD) represent the most frequent hereditary childhood myopathy, affecting approximately 19.8 in 100,000 male births worldwide.”

5.     In the Introduction, “Respiratory function can be affected as well as cardiac function particularly in patients with DMD” also should be improved.

6.     In the introduction, “Because heart and lung are localized in the thorax, intra-thoracic pressure directly affects the heart function, the patient being on spontaneous breathing or on mechanical ventilation.” also should be improved.

… …

Author Response

In this review, FAYSSOIL et al. report the direct cyclic effects of mechanical ventilation on right and left ventricular systolic and diastolic function in DMD patients. I have several significant concerns.

Major Comments

 It is well known that DMD patients displayed cardiomyopathy in late life. Do you have the data showing heart defects in DMD patients?

>>> We thank the reviewer for these comments.  In the Case 1, the DMD patient was on ACE inhibitors because of previous left ventricular systolic dysfunction (left ventricular ejection fraction 45%). In the Case 2, the DMD patient was treated with ACE inhibitors in the context of cardioprotective drugs; the previous LVEF was normal. This was added in the manuscript.

  1. For each patient, can you provide all the five parameters (left ventricular dimensions, LVOT pulsed Doppler flow, Right ventricular (RV) systolic and diastolic tissue Doppler peak velocities, RV diastolic inflow pattern, Right ventricular (RV) systolic and diastolic tissue Doppler peak velocities).

>>> we thank the reviewer for these comments. A table summarized Doppler echocardiographic data available was added in the revised version of the manuscript

  1. The manuscript is clear, but there are many grammatical errors. I suggest that editing for academic English is necessary and would significantly improve the communication of the findings of this paper.

>>>  The spelling and grammar of this new version of the  manuscript was improved

Minor concerns:

  1. In Abstracts, “We report case series of direct cyclic effects of mechanical ventilation on right and left ventricular systolic and diastolic function.” should be changed into ” We report the direct cyclic effects of mechanical ventilation on right and left ventricular systolic and diastolic function in serial cases.”

>>> We thank the review for this suggestion. Modifications were made in the revised manuscript.

  1. In Abstracts, “These cases highlight the importance of performing echocardiography Doppler in patients with neuromuscular disorders «should be changed into “These cases highlight the importance of performing Doppler echocardiography in patients with neuromuscular disorders”

>>> The abstract was modified in this new version of the manuscript, according to the reviewer ‘s suggestion.

  1. In the Introduction, “Neuromuscular disorders are a group of hereditary and acquired neurological diseases with skeletal disability.” should be changed into “Neuromuscular disorders are acquired or inherited (genetic) conditions that affect the peripheral nervous system and skeletal muscles.”

>>> We thank the review for this suggestion. Modifications were made in the revised manuscript.

  1. In the Introduction, “Duchenne muscle dystrophy (DMD) represent the most frequent hereditary childhood myopathy (global DMD birth prevalence: 19.8/100,000 live male births)” should be changed into “Duchenne muscle dystrophy (DMD) represent the most frequent hereditary childhood myopathy, affecting approximately 19.8 in 100,000 male births worldwide.”

>>> We thank the review for this suggestion. These  modifications were now included.

  1. In the Introduction, “Respiratory function can be affected as well as cardiac function particularly in patients with DMD” also should be improved.

>>> We provided a new sentence in the new version of the manuscript.

  1. In the introduction, “Because heart and lung are localized in the thorax, intra-thoracic pressure directly affects the heart function, the patient being on spontaneous breathing or on mechanical ventilation.” also should be improved.

>>> We modified and provide a new sentence in the revised version of the manuscript.

Reviewer 2 Report

Fayssoil et al. reported in a case series the influence of mechanical ventilation on heart function in neuromuscular patients. Since the heart and lungs are located in the chest, intrathoracic pressure can directly influence cardiac function. The cases are reported comprehensively. The discussion is well argued. 

Author Response

 Fayssoil et al. reported in a case series the influence of mechanical ventilation on heart function in neuromuscular patients. Since the heart and lungs are located in the chest, intrathoracic pressure can directly influence cardiac function. The cases are reported comprehensively. The discussion is well argued. 

>>> We thank the reviewer for these positive comments.

Round 2

Reviewer 1 Report

The revised version has already had obvious improvements in language and content presentation.